# Microplastics, Endocrine Disruptors, and Oxidative Stress: Mechanisms and Health Implications

**DOI:** 10.3390/ijms27010399

**Published:** 2025-12-30

**Authors:** Kalman Kovacs, Jozsef Bodis, Reka A. Vass

**Affiliations:** 1Department of Obstetrics and Gynecology, Medical School, University of Pécs, 7624 Pécs, Hungary; bodis.jozsef@pte.hu (J.B.); vass.reka@pte.hu (R.A.V.); 2MTA-PTE Human Reproduction Scientific Research Group, University of Pécs, 7624 Pécs, Hungary; 3National Laboratory on Human Reproduction, University of Pécs, 7624 Pécs, Hungary

**Keywords:** microplastics, nanoplastics, oxidative stress, endocrine disruptors, ROS, mitochondrial dysfunction, bioaccumulation, reproductive toxicity

## Abstract

Microplastics and nanoplastics (<5 mm and <1 μm, respectively) are emerging contaminants now ubiquitous across environmental matrices and increasingly recognized for their impacts on human health. These particles commonly adsorb or contain endocrine-disrupting chemicals—such as bisphenol-A and phthalate additives—that together trigger complex biological responses. This review examines the central role of oxidative stress in mediating the toxicity of microplastics and associated endocrine disruptors across multiple organ systems. We discuss mechanisms including cellular uptake, reactive oxygen species generation, mitochondrial dysfunction, impairment of antioxidant defenses, and activation of key signaling pathways. Organ-specific effects on reproductive health, cardiovascular function, hepatic metabolism, gut barrier integrity, and neurological systems are highlighted. Current evidence strongly supports oxidative stress as a pivotal mechanism linking microplastic exposure to systemic toxicity, underscoring important implications for public health policy and clinical intervention strategies.

## 1. Introduction

Microplastics (MP)—plastic fragments less than 5 mm in diameter, including nanoscale particles under 1 µm—have emerged as pervasive environmental pollutants detected across terrestrial, aquatic, and atmospheric systems. Human exposure is now well established, with microplastics identified in the placenta, blood, lung tissue, and breast milk [1,2,3,4]. These particles frequently adsorb or incorporate endocrine-disrupting chemicals (EDCs) such as bisphenol A (BPA) and phthalates, which can leach readily from polymer matrices. Acting as vectors, microplastics can also accumulate additional environmental contaminants, including heavy metals and persistent organic pollutants, thereby amplifying their toxic potential [5,6,7]. Nanoplastics (NP), due to their small size and high surface reactivity, are particularly concerning, as they have demonstrated the ability to cross critical biological barriers—including the intestinal epithelium, placental barrier, and blood–brain barrier—raising concerns about systemic distribution and long-term bioaccumulation [8,9].

Among the diverse biological effects associated with microplastic exposure, oxidative stress has emerged as a central and unifying mechanism of toxicity. Excessive generation of reactive oxygen species (ROS), combined with impairment of endogenous antioxidant defenses, can initiate a cascade of cellular dysfunctions, including lipid peroxidation, mitochondrial damage, inflammation, and genomic instability. Importantly, many microplastic-associated EDCs—including BPA and phthalates—can independently induce oxidative stress and disrupt hormonal homeostasis, suggesting potential synergistic or additive effects that may exacerbate microplastic-induced toxicity. Additionally, nanoparticles and polymer degradation products may further intensify redox imbalance through catalytic surface reactions and altered mitochondrial electron flow.

Given the increasing detection of microplastics in human tissues and their persistence in the environment, understanding the molecular interplay between microplastics, EDCs, and oxidative stress is critical. This is particularly relevant because oxidative stress not only contributes to organ-specific toxicity but also plays a mechanistic role in endocrine disruption, metabolic dysfunction, immune dysregulation, and reproductive impairment [10,11,12]. Despite growing evidence, the pathways linking microplastic exposure to oxidative and endocrine disturbance remain incompletely understood. Therefore, a comprehensive synthesis of current findings is urgently needed to clarify mechanisms, identify biomarkers of exposure and effect, and guide future research toward assessing human health risks and developing targeted interventions.

Microplastics—plastic fragments less than 5 mm in diameter, including nanoscale particles under 1 µm—have emerged as pervasive environmental pollutants detected across terrestrial, aquatic, and atmospheric systems. Human exposure is now well established, with microplastics identified in the placenta, blood, lung tissue, and breast milk [1,2,3,4]. These particles frequently adsorb or incorporate EDCs such as BPA and phthalates, which can leach readily from polymer matrices. Acting as vectors, microplastics can also accumulate additional environmental contaminants, including heavy metals and persistent organic pollutants, thereby amplifying toxic potential [5,6,7]. NPs, due to their small size and high surface reactivity, are particularly concerning as they have demonstrated the ability to cross critical biological barriers, including the intestinal epithelium, placental barrier, and blood–brain barrier [8,9].

Among the diverse biological effects linked to microplastic exposure, oxidative stress has emerged as a central mechanism of toxicity. Excessive generation of ROS and the concomitant impairment of antioxidant defenses can lead to lipid peroxidation, mitochondrial dysfunction, inflammation, and genomic instability. Notably, many associated EDCs—including BPA and phthalates—can independently induce oxidative stress and disrupt hormonal homeostasis, suggesting potential synergistic or additive effects when combined with microplastic exposure. Given their environmental persistence and documented presence in human tissues, elucidating how microplastics and their chemical constituents trigger oxidative stress is essential [10,11,12]. To address the possibility that oxidative stress is a downstream consequence rather than a primary driver of microplastic-induced toxicity, several studies have employed oxidative stress inhibition approaches. Experimental evidence shows that pharmacological antioxidants, particularly N-acetylcysteine (NAC), significantly attenuate micro- and nanoplastic–induced reactive oxygen species generation, lipid peroxidation, mitochondrial dysfunction, and inflammatory responses, thereby reducing cellular injury and apoptosis [13,14]. Importantly, NAC intervention has been shown to partially or completely restore antioxidant enzyme activity and suppress activation of redox-sensitive pathways such as NF-κB and MAPKs following microplastic exposure, supporting a causal role for oxidative stress in mediating toxicity rather than a purely secondary effect [14,15]. These findings provide mechanistic validation that oxidative stress functions as an upstream trigger of cellular damage in microplastic exposure models and highlight antioxidant modulation as a useful experimental strategy to confirm causality.

MPs/NPs and plastic additives (e.g., bisphenols and phthalates) are distinct exposure agents—MPs/NPs are particulate polymers, whereas additives are low-molecular-weight chemicals blended into plastics—but they are mechanistically linked because many additives are not covalently bound to the polymer matrix and can therefore migrate/leach from plastics and from fragmented MPs/NPs [16,17]. Fragmentation into smaller particles can increase release potential by increasing surface area and shortening diffusion distances, while environmental aging (e.g., UV/oxidation) can enhance surface cracking and chemical mobility [17]. In vivo, desorption remains plausible because physiological media (gastric acidity, bile salts/lipids in the intestine, proteins forming a “corona,” and pulmonary surfactants) can promote partitioning of hydrophobic additives into biological phases, enabling bioavailability even after ingestion or inhalation of MPs/NPs [17]. Once released, endocrine-active additives can interact with hormone receptors and endocrine pathways, providing a plausible route to endocrine disruption and downstream hormonal imbalance [16,18]. However, the quantitative contribution of MP/NP-bound additives to overall internal BPA/phthalate dose relative to other dominant sources (e.g., food-contact materials, dust, consumer products) is still uncertain, supporting the need for mass-balance and tracer-based studies under environmentally relevant chronic low-dose conditions [17].

Across polymer types, microplastics share common downstream toxicological pathways (e.g., oxidative stress, inflammation, and mitochondrial dysfunction), but the most important differences arise from (i) additive/leachate chemistry, (ii) surface–biomolecule interactions (protein corona), and (iii) aging-dependent changes in surface reactivity and contaminant binding. PVC microplastics are often distinguished by their high plasticizer content, and multiple studies show that endocrine-active additives (notably phthalates and sometimes BPA) can leach/desorb from PVC microplastics, with weathering/photoaging further increasing leaching rates—supporting a stronger “chemical-driven” component to toxicity compared with more additive-lean polymers [19,20,21,22]. In contrast, relatively inert PE/PP particles may show weaker intrinsic reactivity but can still contribute substantially via physical irritation (size/shape-dependent) and by acting as sorbents for hydrophobic pollutants, with polymer- and aging-dependent differences in adsorption capacity that can shift observed effects toward a “vector” mechanism [8,23,24]. PS is frequently used in toxicology studies and tends to show pronounced cell interactions that are strongly modulated by protein corona composition, which can alter uptake routes and inflammatory signaling—highlighting that surface chemistry (not only the polymer backbone) drives potency differences [25,26]. Finally, PET (often present as fibers) can differ in airway deposition/clearance due to morphology, and it also forms dynamic coronas in gastrointestinal conditions that can modify surface properties and bio-interactions, underscoring how “polymer type effects” often reflect a combination of formulation, aging, and corona-mediated biology rather than a single polymer-specific pathway [23,27].

This review aims to synthesize current evidence on how microplastics and associated endocrine-disrupting chemicals contribute to oxidative stress and disrupt endocrine signaling, highlighting the molecular pathways that link exposure to cellular damage and hormonal imbalance. It further evaluates the resulting multisystem health implications—particularly for metabolic, reproductive, cardiovascular, and neurobiological outcomes—and identifies key research gaps needed to improve human risk assessment.

## 2. Presence of Micro- and Nanoplastics in Human Tissues

Recent analytical advances have provided direct evidence that MPs and NPs are present in multiple human tissues. Using pyrolysis–GC/MS and Raman microspectroscopy, MPs have been detected in human whole blood, confirming that ingested or inhaled particles can cross epithelial barriers and enter systemic circulation [2]. The first demonstration of MPs in human placentas showed particles embedded on both the maternal and fetal sides, indicating potential transplacental transfer during pregnancy [1]. Subsequent work has identified MPs in lung tissue, where they were found in 11 of 13 surgical specimens, supporting inhalation as a major exposure route [28]. MPs have also been detected in human breast milk, suggesting early-life exposure in infants [29]. More recently, postmortem analyses have confirmed MPs and NPs in multiple organs, including the liver, kidney, thyroid gland, and even the human brain, which in one study showed higher concentrations than other tissues examined [30,31]. Autopsy-based studies also report the presence of MPs in cardiovascular structures, including carotid artery atherosclerotic plaques, where their presence was associated with a higher risk of major cardiovascular events [32]. Taken together, these findings demonstrate that MPs are not restricted to the gastrointestinal tract but can translocate through biological barriers, circulate in the bloodstream, and accumulate in diverse organs. Their widespread presence across human tissues raises significant concerns about long-term health implications, particularly given their links to oxidative stress, inflammation, and endocrine disruption [33] (Table 1). Although daily human exposure to microplastics is increasingly documented, the absence of standardized exposure metrics and human-relevant toxicity thresholds currently precludes a direct quantitative correlation, particularly under conditions of chronic low-dose exposure and cumulative bioaccumulation [17].

## 3. Impact of Microplastics on Multiple Organ Systems

MPs/NPs are increasingly recognized as systemic toxicants rather than localized environmental contaminants. Due to their small size, large surface area, and ability to carry EDCs, MPs/NPs can cross biological barriers, accumulate in tissues, and disrupt cellular signaling across multiple organ systems [2,17]. Experimental and emerging human evidence indicates that oxidative stress, inflammation, endocrine disruption, mitochondrial dysfunction, and altered intercellular signaling are shared mechanisms underlying their multisystem toxicity.

### 3.1. Cardiovascular System

Growing evidence suggests that MPs/NPs adversely affect cardiovascular structure and function. Following inhalation or ingestion, small plastic particles can translocate into systemic circulation and interact directly with vascular endothelial cells [3]. In vitro and in vivo studies demonstrate that MP exposure induces endothelial dysfunction characterized by increased reactive oxygen species (ROS) production, reduced nitric oxide bioavailability, and impaired vasodilation [4]. Oxidative stress-driven activation of NF-κB and MAPK signaling promotes vascular inflammation, increasing expression of pro-inflammatory cytokines such as TNF-α and IL-6, which are key contributors to atherogenesis [5].

Recent animal studies further indicate that chronic MP exposure accelerates lipid accumulation, macrophage infiltration, and plaque instability in vascular tissues, suggesting a potential role in atherosclerosis progression [6]. Notably, MPs have been detected in human blood and atherosclerotic plaques, providing direct evidence of cardiovascular exposure and raising concerns about long-term cardiometabolic risk [2,7]. In addition, MP-induced endocrine disruption, particularly altered lipid and glucose metabolism, may indirectly exacerbate cardiovascular disease risk through metabolic dysregulation.

### 3.2. Nervous System

The nervous system is particularly vulnerable to MP/NP exposure due to the ability of nanoplastics to cross the blood–brain barrier and accumulate in brain tissue [44]. Animal studies have detected MPs/NPs in the cortex, hippocampus, and olfactory bulb, regions critical for cognition, memory, and sensory processing [45]. Neurotoxicity is largely mediated by oxidative stress, mitochondrial dysfunction, and neuroinflammation, with microglial activation and elevated levels of IL-1β and TNF-α commonly reported following exposure [13].

MP-induced disruption of calcium signaling and mitochondrial membrane potential further amplifies ROS production, leading to synaptic damage and neuronal apoptosis [46]. Neurotransmitter systems are also affected, with reported alterations in dopamine, acetylcholine, and glutamate pathways, resulting in behavioral changes such as anxiety-like behavior, impaired learning, and memory deficits in animal models [47]. Of particular concern is early-life exposure, as prenatal and neonatal MP exposure has been linked to abnormal neurodevelopment, altered synaptic plasticity, and long-lasting cognitive impairment [47]. These findings raise important questions regarding the potential contribution of MPs/NPs to neurodevelopmental and neurodegenerative disorders in humans.

### 3.3. Reproductive System

The reproductive system represents one of the most consistently affected targets of MP/NP exposure. MPs act as both sources and carriers of EDCs, including bisphenols and phthalates, which interfere with hormone receptor signaling and steroidogenesis [16,18]. In male reproductive systems, MP exposure has been associated with reduced sperm count, impaired sperm motility, testicular oxidative stress, and disruption of testosterone synthesis through altered expression of steroidogenic enzymes such as CYP11A1 and CYP17A1 [13].

In females, MPs/NPs accumulate in ovarian and uterine tissues, leading to follicular atresia, altered estrous cycles, and disrupted estrogen and progesterone balance [16,18,48]. Placental accumulation of MPs has been reported in both animal models and humans, raising concerns about impaired placental function, fetal hormone exposure, and developmental programming effects [1]. Importantly, several studies demonstrate dose- and time-dependent accumulation of MPs/NPs in reproductive organs, suggesting that chronic low-level exposure may result in progressive reproductive toxicity even in the absence of acute effects [48,49].

## 4. Mechanisms of Oxidative Stress Induced by Microplastics and EDCs

### 4.1. ROS Generation and Antioxidant Defense Impairment

When cells are exposed to microplastics, they rapidly begin generating excessive amounts of ROS. Although some ROS arise from chemical interactions at the particle surface, the majority are produced intracellularly. Once micro- or nanoplastics enter the cell, they can accumulate in mitochondria and other organelles, where they interfere with normal electron transport and disrupt redox homeostasis. Studies show, for example, that polystyrene microplastics induce dose- and concentration-dependent increases in ROS across multiple cell types and animal models [1,2,3,4]. Mechanistically, nanoplastics and microplastics impair the mitochondrial electron transport chain, promoting electron leakage and the formation of superoxide and other reactive species [3,4,5]. In parallel, both the particles and associated chemicals—such as bisphenol A (BPA)—can activate ROS-producing enzymes like NADPH oxidase, further amplifying oxidative stress [6,7].

Under mild oxidative stress, cells attempt to restore balance by upregulating antioxidant defenses, largely through the Nrf2–Keap1 signaling pathway. Under basal conditions, Keap1 retains Nrf2 in the cytoplasm and targets it for degradation. When ROS oxidize critical cysteine residues in Keap1, Nrf2 is released, translocates to the nucleus, and induces the transcription of antioxidant and cytoprotective genes [8,9]. These include genes encoding key enzymes such as superoxide dismutase (SOD), catalase (CAT), and glutathione peroxidase (GPx) [8,10]. Low-level microplastic exposure can trigger this transient protective response, reflected by an initial increase in antioxidant enzyme activities and glutathione-related defenses [4,10,11]. When ROS generation becomes excessive or sustained, however, this defense system is progressively compromised. Antioxidant enzymes may be inactivated or depleted, and intracellular glutathione pools become oxidized, leaving cells unable to neutralize accumulating ROS [11,12]. As oxidative stress intensifies, ROS begin to damage essential cellular components, including lipids, proteins, and DNA. Lipid peroxidation—one major consequence—produces reactive aldehydes such as malondialdehyde, which is consistently elevated in tissues and organs exposed to microplastics [11,12]. Experimental evidence consistently demonstrates that micro- and nanoplastic (MP/NP) exposure induces lipid peroxidation in a dose-dependent manner, with increasing concentrations and longer exposure durations leading to progressive elevations in malondialdehyde (MDA) and other lipid oxidation products, alongside depletion of antioxidant defenses such as superoxide dismutase and glutathione peroxidase [6,14,48]. Notably, nanoplastics often elicit stronger lipid peroxidation responses than larger microplastics at comparable mass doses, reflecting enhanced cellular uptake, greater surface reactivity, and increased mitochondrial dysfunction [46]. In vivo biodistribution studies further reveal dose- and time-dependent accumulation of MPs/NPs in multiple organs following oral or inhalation exposure, with the highest burdens typically observed in the gastrointestinal tract and liver, followed by the kidney, lung, spleen, heart, brain, and reproductive organs [47,48,49]. Importantly, nanoplastics demonstrate a greater ability to cross biological barriers, including the blood–brain and placental barriers, raising concerns about neurodevelopmental and reproductive toxicity under chronic low-dose exposure conditions [46,47]. Together, these findings indicate that MP/NP toxicity is strongly influenced by exposure dose, particle size, and cumulative tissue accumulation, with lipid peroxidation serving as a mechanistic link between particle burden and organ-specific oxidative injury. Similarly, microplastic-induced ROS can cause DNA strand breaks and oxidative base modifications, reflected in increased levels of 8-hydroxy-2′-deoxyguanosine (8-OHdG) and positive comet assay results [28,29]. In summary, microplastic exposure leads to chronic oxidative stress driven by excessive ROS production and weakened antioxidant defenses—a combination that sets the stage for widespread cellular dysfunction [4,5].

### 4.2. Mitochondrial Dysfunction and Cell Death Pathways

Mitochondria are especially vulnerable to the oxidative damage induced by MPs and NPs. As the cell’s primary energy producers and a major endogenous source of ROS, mitochondria can undergo severe functional decline when microplastics accumulate intracellularly. Electron microscopy studies have confirmed that both nano- and microplastics enter cells and localize to mitochondrial compartments, provoking mitochondrial membrane depolarization, structural disruption, and markedly reduced ATP output. For instance, one study found that human fibroblasts exposed to nanoscale polystyrene particles exhibited mitochondrial swelling, loss of membrane potential, diminished ATP production, and cytochrome c leakage—hallmark features of oxidative stress-driven apoptosis [49,50].

This mitochondrial compromise is tightly linked to ROS overproduction: disruptions of the tricarboxylic acid cycle and oxidative phosphorylation by microplastics lead to enhanced electron leakage, which generates excess ROS and further impairs mitochondrial enzyme function [51,52]. As oxidative injury accumulates, cells frequently initiate regulated cell-death pathways. Prolonged exposure to microplastics has been associated not only with mitochondrial-mediated apoptosis (via cytochrome c release and caspase activation) but also with non-apoptotic death modalities such as necroptosis and ferroptosis under conditions of pronounced ROS elevation [53,54,55].

Beyond mitochondria, microplastic-induced oxidative stress can also provoke endoplasmic-reticulum stress and activate the unfolded protein response. In ovarian cells of animals exposed to microplastics, activation of the PKR-like ER Kinase (PERK)–eukaryotic Initiation Factor 2 alpha (eIF2α)—Activating Transcription Factor 4 (ATF4)–C/EBP-Homologous Protein (CHOP) axis has been observed, suggesting that unresolved ER stress—likely driven by ROS and calcium dysregulation—contributes to organ-specific cellular damage [56,57,58,59]. Collectively, mitochondrial dysfunction and ER stress form an interlinked cascade of deleterious events rooted in the oxidative burden imposed by microplastics (Figure 1). Calcium signaling plays a critical role in micro- and nanoplastic–induced cellular stress and is closely linked to reactive oxygen species (ROS) generation. Exposure to micro/nanoplastics disrupts intracellular Ca^2+^ homeostasis by promoting Ca^2+^ influx through plasma membrane channels and release from intracellular stores such as the endoplasmic reticulum, leading to cytosolic Ca^2+^ overload [60,61,62]. Elevated Ca^2+^ levels stimulate mitochondrial Ca^2+^ uptake, which impairs mitochondrial membrane potential and enhances mitochondrial ROS production, thereby amplifying oxidative stress [13,63]. In parallel, Ca^2+^-dependent activation of NADPH oxidases further contributes to ROS amplification, creating a feed-forward loop between Ca^2+^ dysregulation and oxidative damage [15]. Collectively, these findings suggest that Ca^2+^ signaling acts as an important upstream modulator of ROS generation and oxidative injury following micro- and nanoplastic exposure.

### 4.3. Contributions of Associated Endocrine Disruptors

The chemical additives associated with microplastics—particularly BPA and phthalates—have independent toxic effects that intensify both oxidative stress and endocrine disruption, thereby amplifying the overall biological impact of microplastics. BPA, a well-characterized synthetic estrogen, can bind to estrogen receptors and other signaling proteins, leading to broad disturbances in cellular homeostasis [18,63,64]. BPA exposure is widely reported to increase intracellular production of ROS and to activate oxidative stress–mediated apoptotic pathways [65,66,67]. In ovarian granulosa cells, for example, BPA activates the G-protein–coupled estrogen receptor (GPER), triggering a calcium influx and subsequent activation of the ASK1–JNK signaling cascade, ultimately resulting in ROS-driven apoptosis [68].

BPA’s pro-oxidant effects are closely linked to mitochondrial dysfunction. It can uncouple oxidative phosphorylation, impair ATP generation, and suppress key antioxidant enzymes, contributing to redox imbalance [66,69]. These mitochondrial and oxidative disruptions have been associated with impaired oocyte maturation and reduced sperm quality in multiple animal models [69,70].

Phthalates—particularly di-(2-ethylhexyl) phthalate and its metabolite mono-(2-ethylhexyl) phthalate —exert similar oxidative effects. Due to their lipophilic nature, phthalates integrate into cellular membranes, promoting lipid peroxidation, elevating ROS production, and depleting intracellular glutathione stores [6,71]. A broad review of animal studies shows that chronic phthalate exposure consistently produces oxidative stress, marked by increased levels of malondialdehyde and 8-OHdG, along with significant reductions in antioxidant enzymes such as superoxide dismutase and glutathione peroxidase [6,71,72]. These oxidative alterations are strongly implicated in reproductive dysfunction, hepatic injury, and metabolic disturbances [61,72].

Additionally, both BPA and phthalates can interfere with mitochondrial electron transport and activate redox-sensitive transcription factors, further contributing to cellular oxidative burden [6,66]. Together, these findings indicate that microplastics and the endocrine-disrupting chemicals they leach act through overlapping mechanisms—direct ROS generation, suppression of antioxidant defenses, mitochondrial impairment, and membrane damage—making oxidative stress a central driver of their toxicity.

### 4.4. Endocrine Implications and Hormonal Disruption

Exposure to microplastics—and particularly to the EDC additives they transport—can profoundly interfere with endocrine regulation. Many plastic additives are well-characterized EDCs capable of mimicking, antagonizing, or dysregulating endogenous hormone functions. BPA, for example, binds to estrogen receptors ERα and ERβ, as well as other hormone-responsive receptors, thereby disrupting downstream signaling and altering gene transcription profiles linked to reproductive and metabolic processes [18,65]. Likewise, phthalates such as di(2-ethylhexyl) phthalate and its bioactive metabolite MEHP interact with nuclear receptors including the peroxisome proliferator-activated receptors (PPARs), leading to dysregulation of lipid metabolism, steroidogenesis, and overall endocrine homeostasis [66].

These receptor-level interactions translate into measurable physiological disturbances. Multiple studies indicate that microplastic exposure disrupts several hormonal axes, prominently the hypothalamic–pituitary–gonadal (HPG) axis. In aquatic models, chronic exposure to polystyrene nanoplastics suppresses HPG-axis activity: reduced expression of gonadotropin-releasing hormone (GnRH) and luteinizing hormone (LH) genes in the brain, decreased circulating sex-steroid concentrations (estradiol and testosterone), and marked reproductive impairment including fewer eggs laid and reduced hatching rates [67]. Similar findings have been documented in mammalian systems. Rodents exposed orally to polystyrene microplastics show significant reductions in key reproductive hormones (follicle- stimulating hormone, LH, and estradiol), along with histopathological damage to ovarian follicles and testicular seminiferous tubules. These effects are dose- and time-dependent, suggesting cumulative endocrine toxicity with sustained microplastic exposure [64].

Microplastic-associated endocrine disruption is not restricted to reproduction. Other endocrine organs are equally susceptible. BPA has been shown to alter thyroid hormone receptor signaling and interfere with enzymes essential for thyroid hormone synthesis and metabolism, resulting in abnormal thyroxine and triiodothyronine levels [8,65]. Human epidemiological studies report correlations between elevated urinary BPA or phthalate metabolites and altered thyroid parameters—including abnormal TSH and thyroxine concentrations—raising concerns about population-level thyroid disruption [72]. Emerging animal evidence also suggests that nanoplastic exposure may influence adrenal function, reflected in altered cortisol levels in fish and corticosterone fluctuations in rodents, although this area warrants further research [53,67].

A recent comprehensive review concluded that MPs and NPs can disrupt endocrine functions across the entire neuroendocrine axis—from central regulators (hypothalamus, pituitary) to peripheral glands (thyroid, adrenal, gonads)—and that many of these effects are mediated, at least in part, through oxidative-stress pathways [3]. Oxidative damage in steroidogenic tissues such as the ovary and testis can impair hormone biosynthesis, while oxidative injury to thyroid follicular cells may reduce hormone production and promote local inflammation. Moreover, endocrine imbalance can itself exacerbate oxidative stress—for instance, estrogen deficiency weakens antioxidant defenses, potentially creating a self-perpetuating cycle of hormonal and oxidative dysfunction [3].

Microplastics induce endocrine disruption and hormonal imbalance through a close and bidirectional interaction with oxidative stress. Exposure to micro- and nanoplastics increases reactive oxygen species production, which directly impairs steroidogenic enzyme activity (e.g., CYP11A1, CYP17A1, aromatase) and alters hormone synthesis in gonadal and adrenal tissues [14,16]. Oxidative stress also modifies hormone receptor structure and signaling sensitivity via lipid peroxidation and redox-dependent post-translational modifications, thereby disrupting estrogen, androgen, thyroid, and glucocorticoid receptor pathways [15,18]. In parallel, endocrine-disrupting chemicals associated with microplastics further exacerbate oxidative stress by impairing mitochondrial function and antioxidant defenses, creating a feed-forward loop between redox imbalance and hormonal dysregulation [14,17]. Together, these interactions provide a mechanistic framework linking microplastic-induced oxidative stress to sustained endocrine disruption and systemic hormonal imbalance.

These disruptions carry significant clinical implications. Endocrine-active microplastics and their associated EDCs may contribute to otherwise unexplained cases of infertility, developmental abnormalities, metabolic dysregulation, and thyroid disease. For reproductive specialists and endocrinologists, this highlights the importance of incorporating environmental exposure screening into routine diagnostic evaluation, especially as microplastic contamination represents an emerging and increasingly unavoidable component of modern environmental health (Figure 2).

### 4.5. Effect of Micro- and Nanonplastics and Endocrine Disruptors on Metabolome

MPs/NPs together with their associated EDCs, have been shown to induce profound alterations in the metabolome, reflecting systemic metabolic dysregulation. Experimental metabolomics studies consistently report dose- and size-dependent disturbances in lipid, amino acid, carbohydrate, and energy metabolism following MP/NP exposure, with nanoplastics generally exerting stronger effects due to enhanced bioavailability and cellular uptake [14,47,48]. Disruption of lipid metabolism is a prominent feature, characterized by altered phospholipids, fatty acids, and sterol intermediates, which is closely linked to oxidative stress–induced lipid peroxidation and impaired mitochondrial β-oxidation [6,47]. In parallel, MP-associated EDCs such as bisphenols and phthalates interfere with nuclear receptor signaling (e.g., PPARs, estrogen and androgen receptors), leading to reprogramming of metabolic pathways involved in glucose homeostasis, insulin sensitivity, and lipid storage [16,18]. Metabolomic analyses further reveal disruptions in amino acid pathways, including branched-chain amino acids and glutamine–glutamate metabolism, suggesting impaired protein turnover and redox balance [14]. Importantly, in vivo studies demonstrate that these metabolomic shifts often coincide with tissue accumulation of MPs/NPs in metabolically active organs such as the liver, intestine, and adipose tissue, supporting a mechanistic link between particle burden, endocrine disruption, and metabolic reprogramming [49,50]. Collectively, these findings indicate that MP/NP exposure, amplified by co-exposure to EDCs, reshapes the metabolome through integrated effects on oxidative stress, hormone signaling, and mitochondrial function, potentially contributing to metabolic disorders and long-term cardiometabolic risk.

## 5. Conclusions

Microplastics and their chemical additives—especially endocrine-disrupting compounds—are increasingly recognized as a significant threat to human health. A central mechanism behind their toxicity is oxidative stress: microplastic exposure promotes excess reactive oxygen species, weakens antioxidant defenses, disrupts mitochondrial function, and triggers lipid, protein, and DNA damage. These processes drive inflammation, cell death, and organ dysfunction. Experimental studies across species consistently link microplastics to neurological, hepatic, immune, and reproductive harm, while early human data suggest similar associations with oxidative stress biomarkers, hormonal imbalance, and immune alterations.

Microplastics have already been detected in human blood, placentas, and respiratory tissues, and real-world exposure is chronic and cumulative. Although many questions remain—such as whether lifelong microplastic exposure contributes to cardiovascular disease, chronic inflammation, or neurodegeneration—the existing evidence highlights an urgent need for focused research. From a public health standpoint, precautionary action is warranted. Reducing plastic pollution and regulating microplastic content in consumer products are immediate priorities. Microplastics exposure may be considered as a potential contributor to reproductive, endocrine, and metabolic disorders. Long-term toxicology and epidemiological studies will be essential for clarifying health risks and identifying susceptible populations. Early findings suggest that antioxidants and anti-inflammatory agents may mitigate microplastic-induced oxidative stress, offering potential avenues for prevention, especially in vulnerable groups. Because oxidative stress is measurable, biomarkers of DNA and lipid oxidation may help monitor exposure and evaluate future therapies (Figure 3).

Despite growing evidence that micro- and nanoplastics induce oxidative stress, endocrine disruption, and multisystem toxicity, most current studies rely on short-term in vitro or animal models using simplified particle types and exposure levels that may not accurately reflect chronic, low-dose human exposure. Human epidemiological data remain limited, and critical knowledge gaps persist regarding dose–response relationships, long-term bioaccumulation, interactions with co-existing environmental pollutants, and susceptibility during sensitive life stages. Future research should prioritize environmentally relevant exposure models, longitudinal human studies, and integrated mechanistic approaches linking oxidative stress, endocrine disruption, and organ-specific outcomes to improve risk assessment and inform effective regulatory strategies.

In summary, microplastics introduce a persistent, modern environmental stressor capable of disrupting endocrine, reproductive, and immune functions largely through oxidative pathways. Recognizing oxidative stress as the core mechanism of microplastic toxicity can help guide scientific investigation, medical awareness, and public health strategies aimed at protecting human health in an increasingly plastic-dependent world.

## Figures and Tables

**Figure 1 ijms-27-00399-f001:**
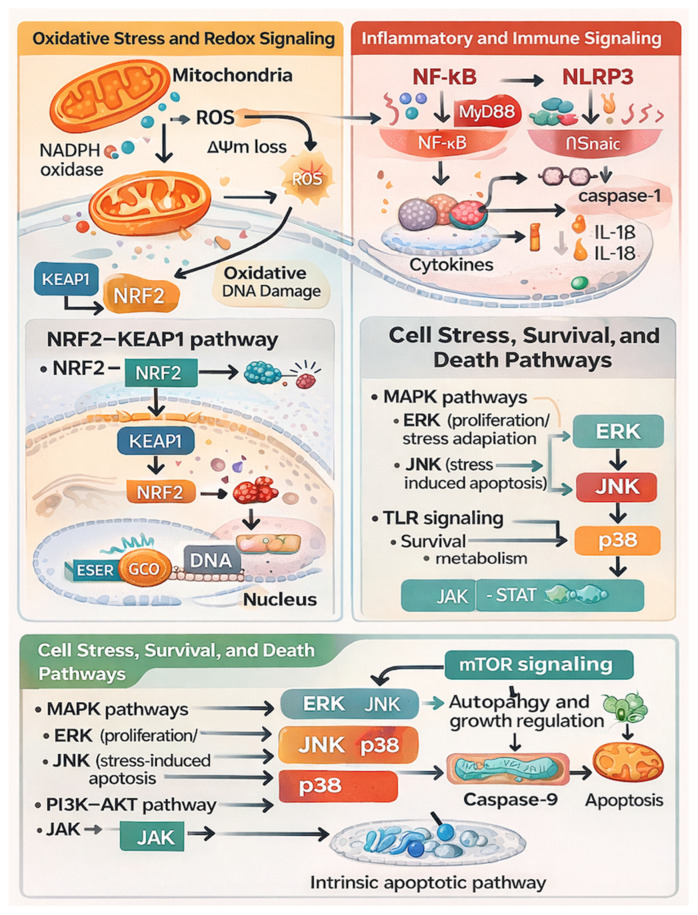
Mechanistic overview of ROS generation and antioxidant response to microplastics.

**Figure 2 ijms-27-00399-f002:**
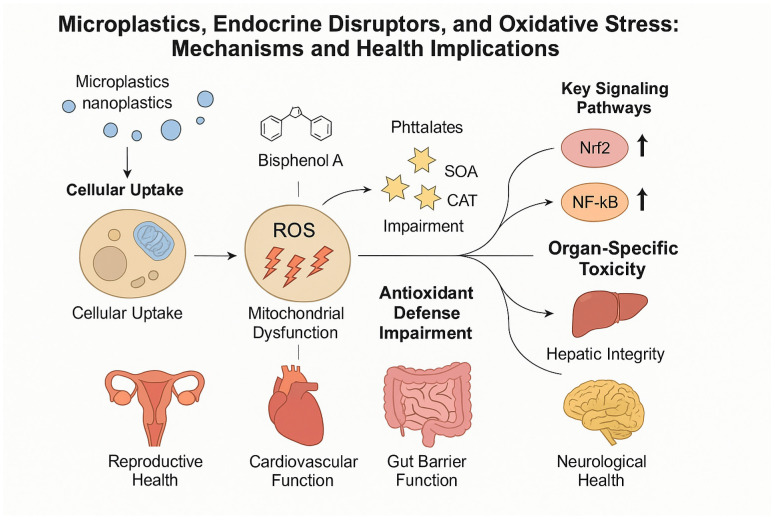
Mechanism and health implications of microplastics, endocrine disruptors, and oxidative stress.

**Figure 3 ijms-27-00399-f003:**
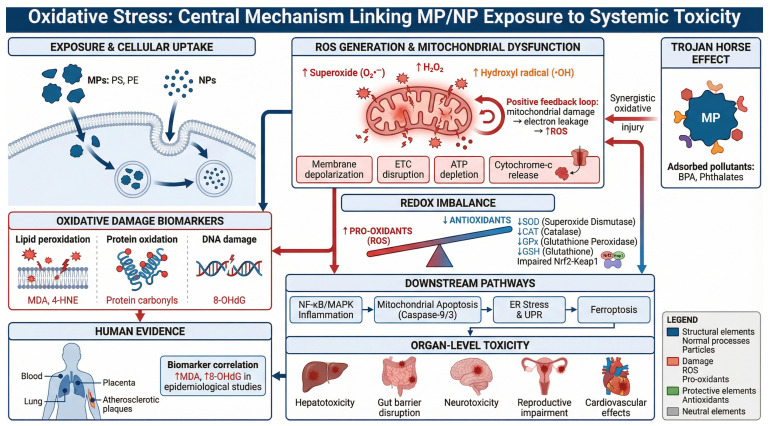
Mechanistic pathways linking microplastic/nanoplastic exposure to oxidative stress and systemic toxicity.

**Table 1 ijms-27-00399-t001:** Proven micro- and nanoplastic components in human tissues and samples.

Human Sample/Tissue	Polymer Types Identified	Particle Size Range	Detection Method	Key Findings	Reference
Whole blood	PET, PE, “polymers of styrene” (e.g., PS-related), PMMA (PP below LOQ)	≥700 nm (method cut-off)	Double-shot Py–GC/MS	Plastics quantified in whole blood of healthy donors; demonstrates systemic internal exposure.	[2]
Placenta	PP (some particles identified by pigment only)	5–10 µm	Raman microspectroscopy	Microplastic fragments found in fetal/maternal sides and membranes in a subset of placentas.	[1]
Placenta, meconium, infant feces, breast milk, infant formula	PE, PP, PS, PU and other common polymers	Mostly 5–500 µm	μFTIR, Raman	Demonstrated perinatal exposure: MPs in placenta and early-life matrices (meconium, infant stool, milk, formula).	[34]
Breast milk	Mainly PE, PVC, PP; pigmented fragments	~2–12 µm (many 4–9 µm)	Raman microspectroscopy	MPs found in 26/34 samples; mostly irregular pigmented fragments.	[29,31]
Lung tissue (autopsy)	Mainly PE, PP	Particles < 5.5 µm; fibers 8.12–16.8 µm	Raman microspectroscopy	Polymeric particles/fibers observed in most samples examined, supporting inhalation-linked deposition.	[35]
Liver (cirrhosis case series), plus kidney and spleen	“Six different MP polymers” (not all specified in abstract)	4–30 µm	Digestion + Nile Red staining/fluorescence microscopy + Raman	MPs detected in cirrhotic tissues but not in controls without underlying liver disease (within detection limits).	[36]
Carotid atherosclerotic plaque (atheroma)	PE; PVC (subset)	Not size-resolved in abstract (micro- and nanoplastics assessed)	Py–GC/MS + stable isotope analysis + electron microscopy	MNPs detected in plaque; presence associated with higher risk of subsequent MI/stroke/death in follow-up cohort.	[32]
Brain, liver, kidney (decedent tissues)	Primarily PE; other polymers present (lesser amounts)	EM verified nanoscale shard-like fragments in brain; full size distribution not limited to >5 µm due to multi-method approach	Py–GC/MS + ATR–FTIR + EM/EDS	Confirms MNPs in organs; brain shows higher PE proportion and higher concentrations; dementia cohort showed greater accumulation and cerebrovascular/immune-cell deposition.	[31]
Bone marrow	PE, PS, PVC, PA66, PP (plus additional polymers by LD-IR)	Majority <100 µm (LD-IR); morphology assessed by SEM	Py–GC/MS + LD-IR + SEM	MPs detected in all bone marrow samples; provides evidence of deep-tissue presence in hematopoietic compartment.	[37]
Testis (human; plus canine comparison)	12 polymer types quantified; PE dominant (PVC/PET showed negative correlations with normalized testis weight in analyses)	Not size-resolved by Py–GC/MS	Py–GC/MS	MPs detected in all tested human testes; exploratory associations with reproductive parameters reported.	[38]
Colon tissue (colectomy specimens)	Polycarbonate, polyamide, PP (subset of analyzed filaments)	Not reported in abstract (reported as filaments/fibers; length characterized in study)	Digestion/filtration + stereo microscopy + micro-FTIR	MPs detected in all colectomy specimens; fibers dominated.	[39]
Saphenous vein tissue (pilot)	Alkyd resin, PVAc, nylon–EVA tie layer (others)	≥5 µm (μFTIR size limitation)	μFTIR	MPs reported in most samples, but an Expression of Concern was issued; interpret cautiously.	[40]
Follicular fluid (human; ART patients)	LD-IR: multiple types (e.g., CPE, fluorosilicone rubber, PVC, etc.); Py–GC/MS confirmed PE/PP/PS/PVC in subset	20–100 µm (LD-IR); Py–GC/MS not size-resolved	LD-IR + Py–GC/MS	MPs detected in follicular fluid; study also tested fluorescent MP beads in vitro and reported impaired oocyte maturation.	[41]
Follicular fluid (human; ART patients)	Polymer ID not reported in abstract (SEM/EDX approach)	<10 µm (mean ~4.48 µm)	SEM + EDX	MPs detected in most samples; reported correlation with FSH in this cohort.	[42]
Cerebrospinal fluid (CSF)	PP, PVC, PE, PS	Not reported in PubMed abstract	Not specified in PubMed abstract (journal article)	CSF microplastics reported; abundance correlated with bottled water frequency and CSF/serum albumin ratio; PE/PVC higher in amyloid-positive group in cohort.	[43]

ART—Assisted reproductive technology; ATR–FTIR—Attenuated total reflectance–Fourier transform infrared spectroscopy; CSF—Cerebrospinal fluid; EDX/EDS—Energy-dispersive X-ray spectroscopy; EM—Electron microscopy; LD-IR/LDIR—Laser direct infrared spectroscopy (chemical imaging); MNPs—Micro- and nanoplastics; MPs—Microplastics; NPs—Nanoplastics; PA—Polyamide (nylon); PA66—Polyamide 66 (nylon 66); PC—Polycarbonate; PE—Polyethylene; PET—Polyethylene terephthalate; PMMA—Polymethyl methacrylate; PP—Polypropylene; PS—Polystyrene; PVAc—Poly(vinyl acetate); PVDF—Poly(vinylidene fluoride); PVC—Poly(vinyl chloride); Py–GC/MS—Pyrolysis–gas chromatography/mass spectrometry; SEM—Scanning electron microscopy.

## Data Availability

No new data were created or analyzed in this study. Data sharing is not applicable to this article.

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
