# Peer review of "Int. J. Mol. Sci.2026, 27(1), 399;https://doi.org/10.3390/ijms27010399"

_ijms, 2025, doi:10.3390/ijms27010399_

Round 1

Reviewer 1 Report

Comments and Suggestions for Authors

This is a timely review, but it is very brief! 

Suggest expanding on the effect on metabolome.

In Fig 1, Are NRF and KEAF1 are the only pathways triggered by micro/nano plastics?

Effect on Ca is mentioned - please expand. Might also be interesting to include Ca signaling and if this has a role in ROS

Include dose effect on lipid peroxidation. Also include accumulation information from in vivo studies (MP/NP accumulation in different organs)

Author Response

Dear Reviewer1,

We thank your time and suggestions to improve our manuscript.

This is a timely review, but it is very brief! 

Suggest expanding on the effect on metabolome.

We have added the paragraph 4.5. about metabolome.

In Fig 1, Are NRF and KEAF1 are the only pathways triggered by micro/nano plastics?

We have revised the figure thoroughly.

Effect on Ca is mentioned - please expand. Might also be interesting to include Ca signaling and if this has a role in ROS

Thank you for the suggestion, we have added a new paragraph about the Ca signaling into section 4.2.

Include dose effect on lipid peroxidation. Also include accumulation information from in vivo studies (MP/NP accumulation in different organs)

We have added more information to section 4.1.

Sincerely,
Dr. Kalman Kovacs med. habil. PhD

on behalf of all authors

Reviewer 2 Report

Comments and Suggestions for Authors

This manuscript focuses on the impact of microplastics and the endocrine disruptors they carry on health through oxidative stress-mediated toxic mechanisms, which holds significant public health implications.  This review has a clear structure, relatively new citations and coherent logic.  It is recommended to accept publication after minor revisions.
1.  Table 1 lists the confirmed microplastic and nanoplastic components in human tissues and samples.  However, the reported ones are far more than just these listed human tissues.  Please supplement them completely.
2.  Both the abstract and conclusion sections have mentioned the impact of microplastics on multiple systems such as the cardiovascular and nervous systems.  However, there is relatively little content related to the nervous and cardiovascular systems in the main text.  It is suggested that appropriate supplementation be made in the corresponding chapters.
3.  In Section 3.4, the mechanism relationship between endocrine disruption and hormonal imbalance related to microplastics was not fully explained.  It is suggested to add the interaction between oxidative stress and endocrine disruption.
4.  The conclusion section can appropriately point out the shortcomings of the current research and the future research directions.
5.  Pay attention to the usage norms of abbreviations.  When they first appear, the full name should be given.
6.  Please standardize the format of references.  Some reference information is still missing, such as references 15, 16, 23, 24, and 27.

Author Response

Dear Reviewer,

We thank your time and suggestions to improve our manuscript.

This manuscript focuses on the impact of microplastics and the endocrine disruptors they carry on health through oxidative stress-mediated toxic mechanisms, which holds significant public health implications.  This review has a clear structure, relatively new citations and coherent logic.  It is recommended to accept publication after minor revisions.

  1. Table 1 lists the confirmed microplastic and nanoplastic components in human tissues and samples.  However, the reported ones are far more than just these listed human tissues.  Please supplement them completely.

Thank you, we added more the details to Table 1.

  1. Both the abstract and conclusion sections have mentioned the impact of microplastics on multiple systems such as the cardiovascular and nervous systems.  However, there is relatively little content related to the nervous and cardiovascular systems in the main text.  It is suggested that appropriate supplementation be made in the corresponding chapters.

Thank you for the suggestion we added a new chapter (3. Impact of Microplastics on Multiple Organ Systems).

  1. In Section 3.4, the mechanism relationship between endocrine disruption and hormonal imbalance related to microplastics was not fully explained.  It is suggested to add the interaction between oxidative stress and endocrine disruption.

We have added a few sentences to section 4.4.

  1. The conclusion section can appropriately point out the shortcomings of the current research and the future research directions.

We have added a few sentences to the conclusion.

  1. Pay attention to the usage norms of abbreviations.  When they first appear, the full name should be given.

Thank you, we have revised the abbreviations.

  1. Please standardize the format of references.  Some reference information is still missing, such as references 15, 16, 23, 24, and 27.

Thank you, we have revised the reference list and corrected it.

Sincerely,
Dr. Kalman Kovacs med. habil. PhD

on behalf of all authors

Reviewer 3 Report

Comments and Suggestions for Authors

This review explores the core role of oxidative stress in mediating the toxicity of microplastics and related endocrine-disrupting chemicals across multiple organ systems, and it holds considerable value. The following issues require attention:

  1. This study presumes oxidative stress to be the core mechanism, yet it fails to rule out the possibility that oxidative stress may be a consequence rather than a cause of cellular damage. It is recommended to cite key evidence from oxidative stress inhibition experiments (e.g., N-acetylcysteine (NAC) intervention) to verify whether such interventions can block microplastic-induced toxicity.
  2. Should the dose-response relationship be taken into account? What is the correlation between the daily human exposure level to microplastics and the toxicity threshold?
  3. Plastic additives (e.g., bisphenol A (BPA), phthalates) and microplastics represent two distinct concepts. It is necessary to clarify whether these chemicals are released from plastics at bioactive doses and whether they can still desorb and exert biological effects in the in vivo environment.
  4. What are the similarities and differences in the toxicological mechanisms among microplastics of different polymer types?
  5. In Lines 280–281 of the conclusion section, the statement “Clinically, microplastic exposure should be considered as a potential contributor to reproductive, endocrine, and metabolic disorders” exceeds the scope of evidence support. To date, no human clinical studies have confirmed a causal association between microplastic exposure and any disease. This wording may mislead clinicians or policymakers.
  6. The studies cited in this article almost uniformly support the notion that “microplastics are toxic”, while negative results or studies debating microplastic toxicity are omitted, which impairs the neutrality of the review.
  7. It is imperative to disclose the policy objectives and policy-oriented background of the funding sources of this study, so as to avoid potential “green science bias”.

Author Response

Dear Reviewer,

We thank your time and suggestions to improve our manuscript.

This review explores the core role of oxidative stress in mediating the toxicity of microplastics and related endocrine-disrupting chemicals across multiple organ systems, and it holds considerable value. The following issues require attention:

  1. This study presumes oxidative stress to be the core mechanism, yet it fails to rule out the possibility that oxidative stress may be a consequence rather than a cause of cellular damage. It is recommended to cite key evidence from oxidative stress inhibition experiments (e.g., N-acetylcysteine (NAC) intervention) to verify whether such interventions can block microplastic-induced toxicity.

We clarified this question in the introduction.

  1. Should the dose-response relationship be taken into account? What is the correlation between the daily human exposure level to microplastics and the toxicity threshold?

We have added a part to section 2.

  1. Plastic additives (e.g., bisphenol A (BPA), phthalates) and microplastics represent two distinct concepts. It is necessary to clarify whether these chemicals are released from plastics at bioactive doses and whether they can still desorb and exert biological effects in the in vivo environment.

We have added information about this into the introduction.

  1. What are the similarities and differences in the toxicological mechanisms among microplastics of different polymer types?

We have added a few details about that into the introduction.

  1. In Lines 280–281 of the conclusion section, the statement “Clinically, microplastic exposure should be considered as a potential contributor to reproductive, endocrine, and metabolic disorders” exceeds the scope of evidence support. To date, no human clinical studies have confirmed a causal association between microplastic exposure and any disease. This wording may mislead clinicians or policymakers.

Thank you, we have corrected this sentence.

  1. The studies cited in this article almost uniformly support the notion that “microplastics are toxic”, while negative results or studies debating microplastic toxicity are omitted, which impairs the neutrality of the review.

Thank you for the feedback, we have added many new paragraphs, hopefully giving a better scope to our paper.

  1. It is imperative to disclose the policy objectives and policy-oriented background of the funding sources of this study, so as to avoid potential “green science bias”.

Funding information was already added to our manuscript.

Sincerely,
Dr. Kalman Kovacs med. habil. PhD

on behalf of all authors

Round 2

Reviewer 3 Report

Comments and Suggestions for Authors

The content added by the author has well responded to my comment and can be published.